# Pitavastatin Exerts Potent Anti-Inflammatory and Immunomodulatory Effects via the Suppression of AP-1 Signal Transduction in Human T Cells

**DOI:** 10.3390/ijms20143534

**Published:** 2019-07-19

**Authors:** Liv Weichien Chen, Chin-Sheng Lin, Min-Chien Tsai, Shao-Fu Shih, Zhu Wei Lim, Sy-Jou Chen, Pi-Fen Tsui, Ling-Jun Ho, Jenn-Haung Lai, Jun-Ting Liou

**Affiliations:** 1Division of Cardiology, Department of Medicine, Tri-Service General Hospital, National Defense Medical Center, Taipei 11490, Taiwan; 2Department of Physiology and Biophysics, Graduate Institute of Physiology, National Defense Medical Center, Taipei 11490, Taiwan; 3School of Medicine, National Defense Medical Center, Taipei 11490, Taiwan; 4Department of Emergency Medicine, Tri-Service General Hospital, National Defense Medical Center, Taipei 11490, Taiwan; 5Graduate Institute of Life Sciences, National Defense Medical Center, Taipei 11490, Taiwan; 6Institute of Cellular and System Medicine, National Health Research Institute, Zhunan, Miaoli 35053, Taiwan; 7Division of Allergy, Immunology and Rheumatology, Department of Internal Medicine, Chang Gung Memorial Hospital, Chang Gung University, Gueishan, Taoyuan 33305, Taiwan

**Keywords:** Atherosclerosis, Inflammation, MAPK pathway, Pitavastatin, T lymphocyte

## Abstract

Statins inhibiting 3-hydroxy-3-methylglutaryl-CoA reductase are the standard treatment for hypercholesterolemia in atherosclerotic cardiovascular disease (ASCVD), mediated by inflammatory reactions within vessel walls. Several studies highlighted the pleiotropic effects of statins beyond their lipid-lowering properties. However, few studies investigated the effects of statins on T cell activation. This study evaluated the immunomodulatory capacities of three common statins, pitavastatin, atorvastatin, and rosuvastatin, in activated human T cells. The enzyme-linked immunosorbent assay (ELISA) and quantitative real time polymerase chain reaction (qRT-PCR) results demonstrated stronger inhibitory effects of pitavastatin on the cytokine production of T cells activated by phorbol 12-myristate 13-acetate (PMA) plus ionomycin, including interleukin (IL)-2, interferon (IFN)-γ, IL-6, and tumor necrosis factor α (TNF-α). Molecular investigations revealed that pitavastatin reduced both activating protein-1 (AP-1) DNA binding and transcriptional activities. Further exploration showed the selectively inhibitory effect of pitavastatin on the signaling pathways of extracellular signal-regulated kinase (ERK) and p38 mitogen-activated protein kinase (MAPK), but not c-Jun N-terminal kinase (JNK). Our findings suggested that pitavastatin might provide additional benefits for treating hypercholesterolemia and ASCVD through its potent immunomodulatory effects on the suppression of ERK/p38/AP-1 signaling in human T cells.

## 1. Introduction

Atherosclerotic cardiovascular disease (ASCVD) with subsequent major vascular events is the leading cause of mortality and morbidity in most developed countries. Hypercholesterolemia, especially with low-density lipoprotein cholesterol (LDL-C), is a major risk factor of ASCVD [1]. Statins, drugs that lower cholesterol through the mechanism of inhibiting 3-hydroxy-3-methylglutaryl-CoA reductase, have demonstrated efficacy and safety in reducing major vascular events in the meta-analyses of many randomized control trials [2], irrespective of baseline risk [3,4], gender [5] or age [6]. Statin therapy is the first-line and fundamental treatment in preventing ASCVD, as suggested by current guidelines [7].

In addition to lipid-lowering effects, statins exhibit systemic anti-inflammatory and immunomodulating properties, which have potent pleiotropic effects in contrast-induced nephropathy, cataract, prostate cancer, and several neurological diseases including head injury, Alzheimer’s and Parkinson’s diseases, and spinal cord injury [8]. Regarding the cellular effects of statins on atherosclerosis, statins upregulate endothelial nitric oxide (NO) synthase (eNOS) and increase NO bioavailability through several mechanisms including rho-associated protein kinase (ROCK) inhibition [9,10]. Additionally, statins increase the effects of interleukin (IL)-18, which inhibits nuclear factor-κB (NFκB) activation, and vascular smooth muscle cell (SMC) migration [11]. Statins display inhibitory effects on the release of the pro-inflammatory cytokines monocyte chemotactic protein-1 (MCP-1), interleukin (IL)-6, and IL-8, and suppress matrix metalloproteinases, in both macrophages and SMCs [12,13]. Specifically, pitavastatin and atorvastatin were proposed to exhibit inhibitory effects on apolipoprotein C3-induced vascular cell adhesion protein 1 (VCAM-1) expression in endothelial cells (ECs) and monocyte adhesion [14]. These studies highlighted statins’ potent atheroprotective effects in addition to their lipid-lowering properties.

Atherosclerosis is a chronic inflammatory disease with lipid dysregulation of the vessel walls [15]. A previous study revealed the aggregation of effector-memory T cells, a sign of activation, in advanced human atherosclerotic plaques [16]. Additionally, the T cells in human atherosclerotic lesions exhibit T helper-1 (Th1) cell-related cytokine secretion, including IL-2, IL-6, interferon (IFN)-γ, and tumor necrosis factor α (TNF-α) [17,18]. These pro-atherogenic cytokines promote the interaction between circulating leukocytes and the endothelium, which further exacerbates the inflammatory progression of artery walls in atherosclerosis [19]. Such evidence points out the mechanistic role of T cells in the process of atherogenesis.

Statins repress oxidized LDL-induced dendritic cell (DC) maturation and suppress T cell activation with reduction of inflammatory cytokine production in T cells from carotid plaques or healthy individuals [20]. Additionally, statins are known to inhibit the differentiation of pro-inflammatory IL-17 helper T cells and promote forkhead box (FOX) P3^+^ CD4^+^ regulatory T cells, which is involved in atherosclerotic plaque stability [21]. In healthy human T cells, statins suppress T cell proliferation and IFN-γ production [22]. However, few studies have evaluated the effects of commonly used statins, such as atorvastatin, rosuvastatin, and pitavastatin, on T cell activation that is relevant to the process of atherosclerosis. In this study, we compared the effects of different statins on human T cell activation and explored the underlying molecular mechanisms. We found that pitavastatin, although showing less lipid-lowering effects than atorvastatin or rosuvastatin, demonstrates higher anti-inflammatory and immunomodulating effects than atorvastatin or rosuvastatin, which is possible through the regulation of extracellular signal-regulated kinase (ERK)/ activating protein-1 (AP-1) and p38/AP-1 signaling pathways.

## 2. Results

### 2.1. Pitavastatin Suppressed Phorbol 12-Myristate 13-Acetate (PMA) Plus Ionomycin-Induced Cytokine Production

First, we evaluated the effects of statins on cytokine production in activated human primary T cells [23]. The IL-2 levels, an indicator of T cell activation, induced by PMA plus ionomycin were significantly inhibited by pitavastatin (32%), atorvastatin (17%), and rosuvastatin (22%) in human primary T cells (Figure 1A). Additionally, the inhibitory effects of pitavastatin, atorvastatin, and rosuvastatin on the secretion of IL-6, IFN-γ, and TNF-α in PMA plus ionomycin-activated T cells were evaluated. The IL-6 levels were inhibited by pitavastatin (59%), atorvastatin (24%), and rosuvastatin (15%); IFN-γ levels were inhibited by pitavastatin (34%), atorvastatin (12%), and rosuvastatin (no significance); and TNF-α levels were inhibited by pitavastatin (61%), atorvastatin (43%), and rosuvastatin (44%) in human primary T cells (Figure 1B–D). To confirm the cellular toxicity of experimental pitavastatin concentrations in human T lymphocytes, 3-[4,5-dimethylthiazol-2-yl]-2,5-diphenyl tetrazolium bromide (MTT) assays and lactate dehydrogenase (LDH) detection were conducted. Pitavastatin at the concentrations of 1, 5, and 10 μM were tested in human T lymphocytes, and no apparent cell toxicities were noted (Figure 1E,F).

### 2.2. Pitavastatin Dose-Dependently Downregulated the mRNA Synthesis of PMA Plus Ionomycin-Induced Pro-Inflammatory Cytokines

We next explored the effects of pitavastatin on PMA plus ionomycin-induced pro-inflammatory cytokine mRNA expression in primary T cells. Consistent with the results in protein levels, pitavastatin dose-dependently downregulated the mRNA expression of IL-2 (33%), IL-6 (25%), IFN-γ (42%), and TNF-α (32%) (Figure 2). The effects of atorvastatin and rosuvastatin on the above-mentioned cytokine mRNA expression were not significant (Figure 2). These data revealed the anti-inflammatory effects of pitavastatin on pro-inflammatory cytokine production in PMA plus ionomycin-activated T cells, which was superior to atorvastatin and rosuvastatin.

### 2.3. Pitavastatin Significantly Inhibited PMA Plus Ionomycin-Induced Mitogen-Activated Protein Kinase (MAPK) Pathways Including ERK and p38

Previous studies suggested that the MAPK and NFκB signaling pathways are involved in T cell activation [24,25]. We further investigated whether pitavastatin regulated the activities of MAPK signaling in PMA plus ionomycin-activated T cells. Pitavastatin significantly suppressed the phosphorylation of ERK and p38, but not c-Jun N-terminal kinase (JNK) (Figure 3A,B). We next examined the effects of pitavastatin on the NFκB pathway. We found that pitavastatin did not affect PMA plus ionomycin-induced phosphorylation of IκBα, which promoted NFκB activation by subunit p65/50 translocation to the nucleus to recall the sustained NFκB transcriptional activity (Figure 3C,D). These data suggest that pitavastatin downregulated the secretion of pro-inflammatory cytokines through MAPK pathways, including ERK and p38, but not JNK and NFκB signaling.

### 2.4. Pitavastatin Downregulated the Transcriptional Activity of AP-1

To further evaluate the molecular mechanisms of pitavastatin underlying the downregulation of T cell activation, we verified the effects of pitavastatin on the activations of MAPK downstream transcription factors, AP-1. We collected nuclear extracts and performed the electrophoresis mobility shift assay (EMSA). The results demonstrated the suppressive effects of pitavastatin on the DNA-binding activity of AP-1 in stimulated T cells (Figure 4A and B). Then, we evaluated the effects of pitavastatin on AP-1 transcriptional activity by transient transfection of the plasmids with AP-1 promoter-encoded luciferase to the 293T cells, which demonstrated that pitavastatin significantly downregulated the luciferase activities in activated 293T cells (Figure 4C).

## 3. Discussion

Statins, such as pitavastatin, perform a central role in the treatment of hypercholesterolemia and ASCVD. Although the pathogenesis of atherosclerosis is concerned with unregulated lipid metabolism, an unfollowed immune response resulting in a sustained chronic inflammation of the arterial wall is addressed in the disease initiation and progression. In our study, we found that pitavastatin was able to suppress the inflammatory response mediated by PMA plus ionomycin in human primary T cells. The underlying molecular mechanism of pitavastatin was through reducing the upstream ERK and p38 kinase activation (Figure 5). These findings highlight the therapeutic potential of pitavastatin in ASCVD.

The guidelines on the management of blood cholesterol call for high-intensity statin therapy, such as atorvastatin or rosuvastatin at higher doses, for patients with clinical ASCVD. However, for those patients with contraindication to high-intensity statins or experience of statin-associated side effects, moderate-intensity statins, like pitavastatin, are also a class 1 recommendation [7]. In a recent randomized superiority trial in Japanese patients with stable coronary artery disease, pitavastatin dose-dependently reduced cardiovascular events [26], similarly to high-intensity statins. A previous study showed that pitavastatin attenuated the progression of atherosclerotic mice, and decreased plaque formation and thrombosis in hypercholesterolemic rabbits [27]. Besides, pitavastatin administration in vivo inhibited the inflammation of the endothelium and leukocyte recruitment in a dorsal air pouch inflammation model of hypercholesterolemic low-density lipoprotein receptors knockout (*Ldlr*^−/−^) mice [14]. Although the development of atherosclerosis was concerned with dysregulated lipid metabolism, the chronic inflammation of vessel walls could accelerate plaque formation, thereby promoting disease progression. The vascular inflammation involved multiple cell accumulation, such as ECs, monocytes, macrophages, SMC, and T cells. Therefore, the pleiotropic effects of pitavastatin may act at multiple points in the continually gathering cascade of events, resulting in atherosclerosis.

The anti-atherogenic effect of pitavastatin has been evaluated in SMC. Pitavastatin suppresses the angiotensin (Ang) II and the platelet-derived growth factor (PDGF)-BB-induced migration and proliferation of SMCs [28]. Besides, as the inflammation occurred in SMCs, pitavastatin inhibited oxidation-induced MCP-1 expression through modulation of the reactive oxygen species (ROS) response [29]. It has been claimed that pitavastatin displays anti-oxidative effects not only in SMCs, but also in ECs [30]. By enhanced eNOS expression, pitavastatin helpfully produced NO, which is essential for cellular responses that repair the function of ECs. In addition, pitavastatin protected ECs from cell death, mediated by the activation of T cells [31]. Pitavastatin also suppressed the inflammatory response in ECs by manipulating inflammation-mediated IL-8 production [32]. Large amounts of immune cells were found in vascular inflammation and plaque formation. Anti-atherogenic effects of pitavastatin have been reported in the downregulation of MCP-1 and regulated on activation, normal T cell expressed and secreted (RANTES) in monocytes. Moreover, pitavastatin suppressed apolipoprotein-induced monocyte adhesion, indicating the inhibitory effects of pitavastatin on monocyte activation [14]. The current study demonstrated the anti-inflammation effects of pitavastatin in T cells. This finding is also consistent with previous studies in which pitavastatin diminished T cell proliferation and IFN-γ expression in activated mouse T cells [22]. Furthermore, pitavastatin ameliorated the severity of experimental autoimmune myocarditis through the inhibition of Th1 and Th17 differentiation in mice [33].

Current data revealed that pitavastatin inhibited the production of IL-2, IL-6, IFN-γ, and TNF-α in activated human primary T cells, indicating an immunomodulatory effect of pitavastatin when facing an inflammatory challenge. Different from the lipophilic nature of atorvastatin and pitavastatin, the lipophobic rosuvastatin performs intracellular effects via membrane-bound transporters, such as organic anion transporting polypeptides (OATP), which does exist in T cells [34,35]. Moreover, a study from Nakagomi et al. suggested that atorvastatin and pitavastatin exhibit distinct effects on inflammation in patients with dyslipidemia [36]. Such evidence partly explains the similar effects of atorvastatin and rosuvastatin on proinflammatory cytokines production in activated T cells, and further supports the specific pleiotropic effects of pitavastatin.

The Th1 cytokines, especially IFN-γ and TNF-α, play an important role in atherosclerosis [37]. The targeting of these cytokines as a therapeutic strategy has been continuously studied. A previous study shows that the administration of recombinant IFN-γ protein provokes severe atherosclerosis in hypercholesterolemic mice [38], whereas the deletion of IFN-γ or its receptor was effective in reducing the disease severity [39]. Besides, obese IFN-γ-deficient mice with significantly reduced TNF-α and MCP-1 in their adipose tissue experienced a decreased the accumulation of inflammatory cells and recovered their glucose tolerance compared to control animals fed with the same diet [40]. In addition, large amounts of IL-6 were detected in atherosclerotic plaques [41]. The previous study on *Ldlr*^−/−^ mice showed that mice who received the IL-6 receptor antibody significantly decreased their atherosclerotic lesions [42]. The inhibitory effect of pitavastatin in IL-6 production was correlated to previous cancer studies [43,44]. Therefore, the immunomodulatory effects of pitavastatin in activated primary T cells strongly support the benefit of pitavastatin for therapeutic application in atherosclerosis. To further evaluate the potential effects of pitavastatin in vivo, the involvement of pitavastatin-treated plasma in a clinical cardiovascular patient sample could be verified.

AP-1, acting as transcriptional factors, are dominantly involved in the synthesis of several proinflammatory cytokines [45,46]. Pitavastatin has been claimed to display beneficial effects on AP-1 inhibition to arrest lysophosphatidylcholine-induced vascular SMC proliferation [47]. The current study revealed that pitavastatin selectively targeted AP-1 DNA binding and transcriptional activity in stimulated T cells and 293T cells, respectively. Our results were consistent with a previous study which showed that ursolic acid suppressed the inflammation-mediated expression of pro-inflammatory cytokines IL-6 and IFNγ through targeting AP-1 signaling in T cells and macrophages [48]. Moreover, the MAPK pathway (the upstream signaling transduction of AP-1 activation) has been widely discussed in relation to multiple inflammatory diseases. Pitavastatin has shown its potency to inhibit IL-1β-induced MAPK activation, of either ERK, p38, or JNK, in human synovial cells [49]. In our study, we demonstrated that treatment with pitavastatin suppressed the PMA plus ionomycin-induced activation of p38 and ERK, but failed to inhibit the phosphorylation of JNK. Although a previous study claimed an effect of pitavastatin in modulating JNK activity, it has been shown that pitavastatin particularly targets the phosphorylation of ERK and p38 MAPK after stimulation with Ang II in vascular SMCs [50]. ERK and p38 are known to affect the downstream molecules, in cooperation with each other, in the MAPK pathway. Therefore, in our investigation, pitavastatin might modulate inflammatory cytokine expression through the ERK/p38-AP-1 cascade. Besides, it is well known that NFκB signaling performs the dominant role in inflammation. Pitavastatin has shown its potential modulatory effects on NFκB activation, activated by TNF-α, in a breast carcinoma cell line [43]. However, Habara et al. show that pitavastatin had no effects on the IL-1β-induced degradation of IκB or the activation of NFκB in hepatocytes [51]. In our study, as in the results for hepatocytes, we showed that pitavastatin did not affect the PMA plus ionomycin-induced activation of the NFκB pathway, neither in DNA-binding activity nor in transcriptional activity (data not shown) [52]. Therefore, we considered that the effects of pitavastatin on NFκB modulation were distinct in different cell types with different stimuli. Additionally, although the current study revealed the potential mechanisms of pitavastatin in activated T cells, the specific targets of pitavastatin should be further evaluated.

In this study, we demonstrated that pitavastatin significantly inhibited the PMA plus ionomycin-induced expression of Th1-related cytokines, including IL-2, IL-6, IFN-γ, and TNF-α, by suppressing the DNA-binding activity of AP-1 and the activation of MAPK pathways (ERK and p38) in human T cells. These findings indicate that pitavastatin may be a promising therapeutic strategy for ASCVD.

## 4. Materials and Methods

### 4.1. Reagents and Antibodies

The purified compound pitavastatin, atorvastatin, and rosuvastatin were obtained from Sigma-Aldrich Chemical Company (St. Louis, MO, USA) and dissolved in dimethyl sulfoxide (DMSO; Sigma-Aldrich Chemical Company). Antibodies against pERK, ERK, pp38, p38, pJNK, JNK, pIκBα, and IκBα were purchased from Cell Signaling (Danvers, MA, USA). The antibodies against tubulin were purchased from Abcam (Cambridge, UK). All other reagents without indications were purchased from Sigma-Aldrich Chemical Company.

### 4.2. Isolation of Primary Human T Cells and Cell Culture

Whole blood buffy coats were collected from healthy volunteers aged 30–60, with prior approval from the Institutional Review Board of the Tri-Service General Hospital (Protocol No. TSGH-2-106-05-096, 24 June 2017). The isolation of human primary T cells from peripheral blood mononuclear cells (PBMCs) were performed as previously described [53]. Purified PBMCs were collected after ficoll hypaque (GE Healthcare; Uppsala, Sweden) gradient separation, and incubated with L243 [anti-DR; American Type Culture Collection (ATCC), Rockville, MD, USA], OKMI (anti-CD11b; ATCC), and LM2 (anti-MacI; ATCC) antibodies at 4 °C for 30 min. The cells were subsequently incubated with magnetic beads that were conjugated with goat anti-mouse IgG (R&D Systems, Minneapolis, MN, USA). The antibody-stained cells were then removed using a magnet. These procedures were repeated, and T cells were obtained with a purity of approximately 90%, as measured by the percentage of CD3^+^ cells through flow cytometry (Beckton Dickinson, Mountain View, CA, USA). The collected T cells were maintained in RPMI-1640 medium (Gibco, New York, USA) supplemented with 10% fetal bovine serum (FBS) and 1% penicillin/streptomycin at 37°C and were used for further experiments. For T cell activation, 5 ng/mL PMA and 1 μM ionomycin (Sigma) were used as stimuli. After stimulation and treatment, the cell pellets or supernatants were collected at the indicated time points for further analysis.

293T cell line was obtained from ATCC (Rockville, MD, USA). The cells were maintained in Dulbecco’s modified Eagle’s medium (DMEM; Gibco, New York, USA) with 10% FBS, 2 mM glutamine, and 1% penicillin/streptomycin at 37C, 5% CO2.

### 4.3. Measurement of Cytotoxicity

To determine the cytotoxicity of pitavastatin, we conducted two assays, namely, 3-[4,5-dimethylthiazol-2-yl]-2,5-diphenyl tetrazolium bromide (MTT) assay and lactate dehydrogenase (LDH) assay (Roche, Indianapolis, IN, USA), according to manufacturer’s instructions. For cell viability detection, MTT assays were performed as previously described [54]. In brief, 2 × 10^5^ T cells in 200 µL of volume were pretreated with pitavastatin for 2 h and incubated in the presence or absence of PMA (5 ng/mL) plus ionomycin (1 μM) for 24 h. Subsequently, 25 μL of MTT (5 mg/mL in H_2_O) was added and the T cells were further incubated at 37 °C for 2 h. The lysis buffer containing 20% sodium dodecyl sulfate and 50% dimethylformamide was added and incubated at 37°C for another 6 h. The amount of dissolved reduced MTT crystals was measured using an enzyme-linked immunosorbent assay (ELISA) reader (Dynatech; Chantilly, VA, USA). For cytotoxicity measurements, the release of LDH has been recognized as an indicator of damage to the plasma membrane and cell death. The cytotoxicity (%) was calculated as ((sample value–medium control)/(high control–medium control)) × 100. The absorbance values of the untreated cell culture supernatants were used as the medium control and an equal quantity of cells treated with 1% Triton X-100 were used as the high control. The sample values were the average absorbance values from triplicate measurements of the cell culture supernatants.

### 4.4. Enzyme-linked Immunosorbent Assay (ELISA)

Cytokine concentrations were determined by Duoset ELISA Kits purchased from R&D Systems (Minneapolis, MN) through the procedure described in Reference [53]. In brief, a 96-well flat-bottom plate was coated with cytokine mAb in phosphate-buffered saline (PBS; pH 7.3) and incubated at room temperature (RT) overnight. The plate was then washed three times with PBS containing 0.05% Tween 20 (PBST). Subsequently, the plate was blocked in PBS with 1% bovine serum albumin, 5% sucrose, and 0.05% NaN_3_ for 1 h. The collected supernatant was further added into each well and incubated for 2 h at RT. After washing with PBST three times, the plate was incubated with biotinylated cytokine detection antibodies for 2 h at RT. Following the repetition of the washing steps, streptavidin-horseradish peroxidase was added, and the plate was incubated at RT for 20 min. Finally, the plate was washed five times, and a substrate solution was added and incubated at RT for another 20 min. The reaction was stopped by adding a stop solution and the cytokine concentrations were determined using a microplate reader (Dynatech).

### 4.5. Quantitative Real-Time Polymerase Chain Reaction (qRT-PCR)

Cellular RNA was isolated using an RNeasy kit (QIAGEN, Hilden, Germany), according to the manufacturer’s instructions [55]. The concentration and purity were determined by a Nanodrop spectrophotometer. Next, 1 μg RNA was reverse transcribed into cDNA by an iScript cDNA synthesis kit (Bio-Rad, Hercules, CA), according to the manufacturer’s protocol. A mixture of 20 ng cDNA, 200 nM primers, and KAPA SYBR FAST kit (Kapa Biosystems; Boston, US) was used in qRT-PCR. To detect the indicated gene expression, 40 cycles of PCR reaction at 95 °C for denaturation and at 60 °C for annealing and extension were monitored on a Roche LightCycler 480 (Roche, Basel, Switzerland). Specific primers for target genes were designed and obtained from National Center for Biotechnology Information (NCBI), the following sequences were used: *GAPDH*, 5′-ATGGGGAAGGTGAAGGTCG-3′ and 5′-TAAAAGCAGCCCTGGTGACC-3′; *IL2*, 5′-TCAAACCTCTGGAGGAAGTGC-3′ and 5′-CATGAATGTTGTTTCAGATCCCTTT-3′; *IL6*, 5′-TCAATATTAGAGTCTCAACCCCCA-3′ and 5′-GAAGGCGCTTGTGGAGAAGG-3′; *IFNG*, 5′-CTGTTACTGCCAGGACCCAT-3′ and 5′-TCTGTCACTCTCCTCTTTCCA-3′; *TNF*, 5′-CCCATGTTGTAGCAAACCCTC-3′ and 5′-TATCTCTCAGCTCCACGCCA-3′. The relative quantification of target mRNA was calculated based on the equation: fold change = 2^−Δ (Δ*C*t)^, where ΔCt = Ct_targeted gene_ − Ct_GAPDH_, and Δ (ΔCt) = ΔCt_stimulated_ − Δ_Ct control_.

### 4.6. Western Blots

Enhanced chemiluminescence Western blotting (Amersham-Pharmacia, Arlington Heights, IL, USA) was operated according to the manufacturer’s instructions, following our previous study [23]. In brief, the collected cells were pelleted and resuspended in a lysis buffer (20 mM 4-(2-hydroxyethyl)-1-piperazineethane-sulfonic acid (HEPES), pH 7.9; 420 mM NaCl; 1.5 mM MgCl_2_; 0.2 mM ethylenediamine tetraacetic acid (EDTA); 25% glycerol; 1 mM dithiothreitol (DTT); 1 mM phenylmethylsulfonyl fluoride (PMSF); and 3.3 μg/mL aprotinin). After vigorous vortexing, the mixture was centrifuged at 14,000× *g* for 20 min. The supernatant was collected and the protein concentration was determined through the Bradford Assay (Bio-Rad, Hercules, CA). Equal amounts of protein were analyzed by 10% sodium dodecyl sulfate polyacrylamide gel and subsequently transferred to a nitrocellulose membrane. For immunoblotting, the nitrocellulose membrane was incubated with Tris buffered saline with Tween-20 (TBST) containing 5% nonfat milk (milk buffer) for 1 h and then blotted with antibodies against indicated protein at 4 °C overnight. After washing with TBST twice for 10 min, the filter was incubated with a secondary antibody for 1 h. The filter was subsequently incubated with the substrate and exposed to X-ray films. The signals were quantified by IMAGEJ software (National Institutes of Health, Bethesda, MD, USA).

### 4.7. Nuclear Extract Preparation

Nuclear extracts were prepared as described in Reference [23]. Briefly, collected cells were incubated with 1 mL of buffer A (10 mM HEPES, pH 7.9; 10 mM KCl; 1.5 mM MgC1_2_; 1 mM DTT; 1 mM PMSF; and 3.3 μg/mL aprotinin) at 4°C with occasional gentle vortexing, for 30 min. To remove the cytoplasmic extract, the swollen cells were then centrifuged at 14,000 rpm for 3 min and the supernatants were discarded. The nuclear pellet was washed with 1 mL of buffer A and centrifuged at 14, 000 rpm for 3 min. Subsequently, the nuclear pellets were incubated with 30 μL of buffer C (20 mM HEPES, pH 7.9; 420 mM NaCl; 1.5 mM MgCl_2_; 0.2 mM EDTA; 25% glycerol; 1 mM DTT; 0.5 mM PMSF; and 3.3 μg/mL aprotinin) at 4°C for 30 min, with occasional vigorous vortexing. After centrifugation at 14,000 rpm for 30 min, the supernatants of the mixture were collected and used as the nuclear extracts.

### 4.8. Electrophoresis Mobility Shift Assay (EMSA)

The EMSA was performed as described previously [23]. Oligonucleotides containing AP-1 or NFκB binding sites were purchased (Promega; Madison, WI, USA), and used as the DNA probes. The DNA probes were radiolabeled with [γ-^32^P] ATP by using T4 kinase (Promega), and further incubated with 5 μg of nuclear extracts and the binding buffer at room temperature for 20 min to enable binding. The final reaction mixture was analyzed using a 6.6% non-denaturing polyacrylamide gel with 0.5 × Tris-borate–EDTA as the electrophoresis buffer.

### 4.9. Transient Transfection and Luciferase Activity Assay

For transient transfections, 293T cells were seeded at 2×10^6^ in a 6-well plate, 1 day before transient transfection. The expression vector containing the AP-1 or NF-κB luciferase reporter constructs was purchased from Promega. The 293T cells were transfected with the mixture of Opti-MEM containing the indicated plasmids and Lipofectamine 2000 reagent (Invitrogen, Carlsbad, Calif., USA). After 6 h of incubation, the medium was replaced with DMEM containing 10% FBS. The cells were allowed to settle at 37 °C for 20 h and seeded equally to different groups for further treatment and stimulation as indicated. To detect the luciferase activity, the cell lysates were lysed and the Luciferase Assay System (Promega) was used according to the manufacturer’s instructions.

### 4.10. Statistical Analysis

Data are represented as median with individual data and the statistical analysis with raw data was interpreted using GraphPad Prism v.6.01 software (GraphPad Software Inc., San Diego, CA, USA). The statistical significance was analyzed by one-way ANOVA with Bonferroni’s post-hoc comparisons, and *p* values less than 0.05 were considered as significant (* *p* < 0.05; ** *p* < 0.01; *** *p* < 0.001; **** *p* < 0.0001).

## Figures and Tables

**Figure 1 ijms-20-03534-f001:**
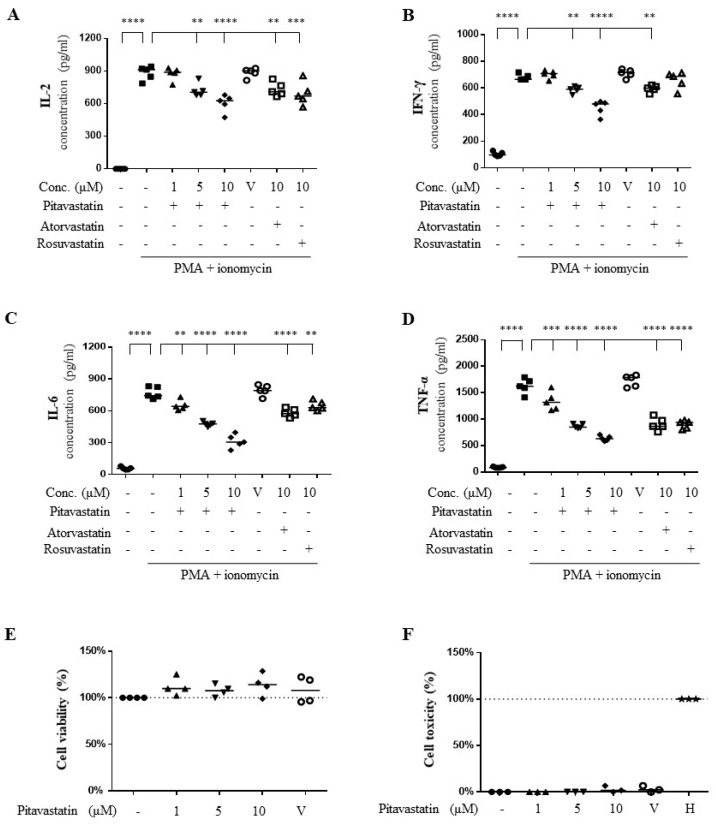
Pitavastatin suppressed pro-inflammatory cytokine production more than atorvastatin and rosuvastatin in phorbol 12-myristate 13-acetate (PMA)+ionomycin-activated primary T cells. (**A**–**D**) Human primary T cells were pretreated with indicated dose of pitavastatin (1, 5, and 10 μM), 10 μM atorvastatin, or 10 μM rosuvastatin, respectively, for 2 h. Following 22 h of PMA (5 ng/mL)+ionomycin (1 μM) stimulation, supernatants were collected for measuring interleukin (IL)-2 (**A**), interferon (IFN)-γ (**B**), IL-6 (**C**), and tumor necrosis factor α (TNF-α) levels (**D**) by enzyme-linked immunosorbent assay (ELISA) methods. (**E**,**F**) Human primary T cells were treated with pitavastatin at the indicated concentrations for 24 h. The cells and cell-cultured supernatants were collected for 3-[4,5-dimethylthiazol-2-yl]-2,5-diphenyl tetrazolium bromide (MTT) assays (**E**) and lactate dehydrogenase (LDH) detection (**F**), respectively. V indicates vehicle control. H indicates the cells treated with 1% Triton X-100 as positive control (100% toxicity). Data were expressed as the median with individual data from 3–5 independent experiments. Significance were represented as ** *p* < 0.01, *** *p* < 0.001, and **** *p* < 0.0001 versus PMA+ionomycin-stimulated group.

**Figure 2 ijms-20-03534-f002:**
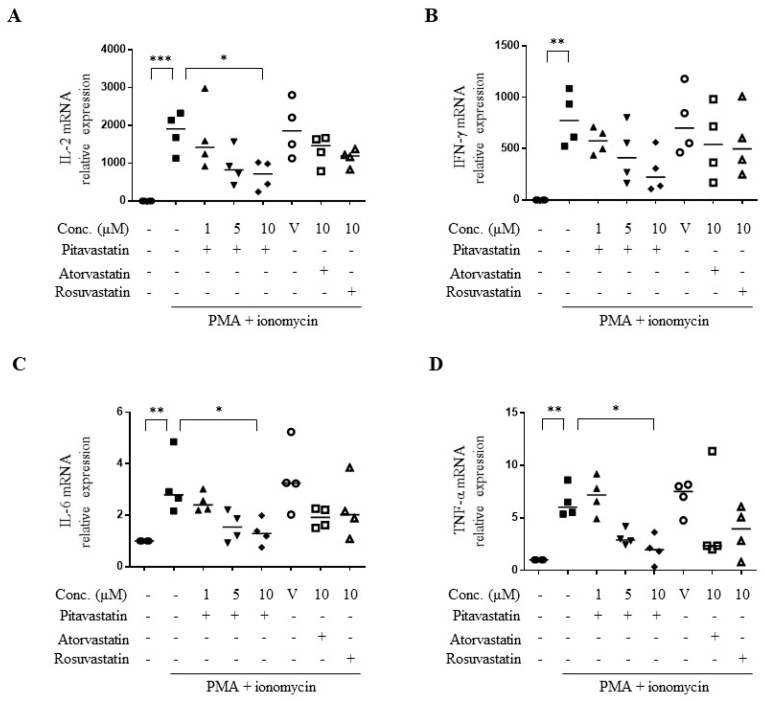
Pitavastatin dose-dependently inhibited *de novo* synthesis of pro-inflammatory cytokines in PMA+ionomycin-activated primary T cells. (**A**–**D**) Human primary T cells were pretreated with indicated dose of pitavastatin, 10 μM atorvastatin, or 10 μM rosuvastatin, respectively, for 2 h. Following 22 h of PMA (5 ng/mL)+ionomycin (1 μM) stimulation, cells were collected for quantitative real time polymerase chain reaction (qRT-PCR) analysis for IL-2 (**A**), IFN-γ (**B**), IL-6 (**C**), TNF-α (**D**) mRNA expression. V indicates vehicle control. Data were expressed as the median with individual data of four independent experiments. Significance were represented as * *p* < 0.05, ** *p* < 0.01, and *** *p* < 0.001 versus PMA+ionomycin-stimulated group).

**Figure 3 ijms-20-03534-f003:**
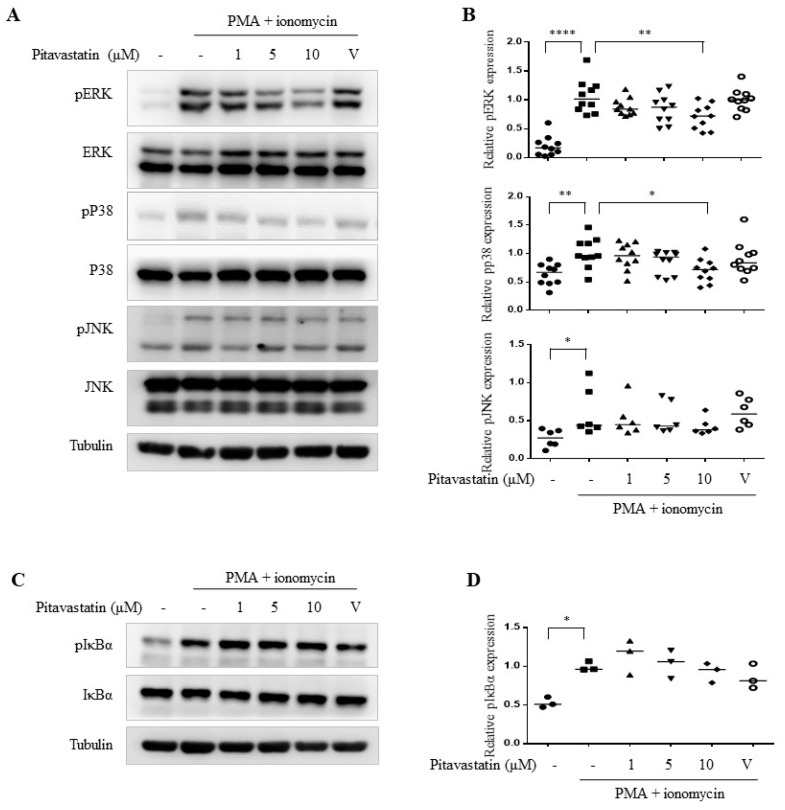
Pitavastatin significantly blocked the activation of both extracellular signal-regulated kinase (ERK) and p38 mitogen-activated protein kinases (MAPKs) in PMA+ionomycin-activated primary T cells. (**A**) Human primary T cells were pretreated with indicated pitavastatin concentrations for 2 h and were subsequently stimulated with PMA (5 ng/mL)+ionomycin (1 μM) for 1 h. Collected cell lysates were analyzed through western blots. Representative data of 6–10 independent experiments were shown. (**B**) Quantitative results of (**A**) were shown. (**C**) Human primary T cells were cultured as (**A**) and the cell lysates were collected for the western blot assay. Representative data of three independent experiments were shown. (**D**) Quantitative results of (**C**) were shown. V indicates vehicle control. Quantitative data (**B**,**D**) were expressed as the median with individual data of 3–10 independent experiments. Significance were represented as * *p* < 0.05, ** *p* < 0.01, and **** *p* < 0.0001 versus PMA+ionomycin-stimulated group).

**Figure 4 ijms-20-03534-f004:**
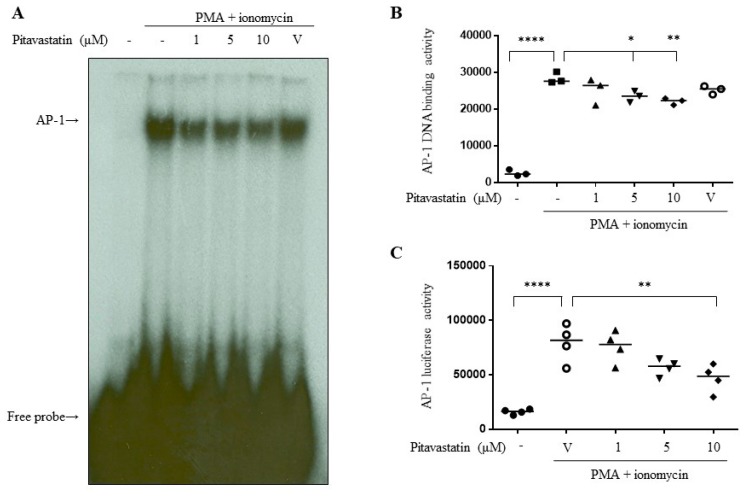
Transcriptional activity of activating protein-1 (AP-1) was blocked by pitavastatin. (**A**) Human primary T cells were pretreated with pitavastatin for 2 h and further stimulated with PMA (5 ng/mL)+ionomycin (1 μM) for 22 h. Collected nuclear protein extracts were analyzed by electrophoresis mobility shift assay (EMSA). Representative data from three experiments were shown. (**B**) Quantitative results of (**A**) were shown. (**C**) 293T cells were transfected with plasmids encoded by the AP-1 promoter. Transfected cells were pretreated with pitavastatin for 2 h and further stimulated with PMA (5 ng/mL)+ionomycin (1 μM) for 22 h. Cell lysates were collected for determining luciferase activity. V indicates vehicle control. Data were expressed as the median with individual data of 3-4 independent experiments. Significance were represented as * *p* < 0.05, ** *p* < 0.01, and **** *p* < 0.0001 versus PMA+ionomycin-stimulated group).

**Figure 5 ijms-20-03534-f005:**
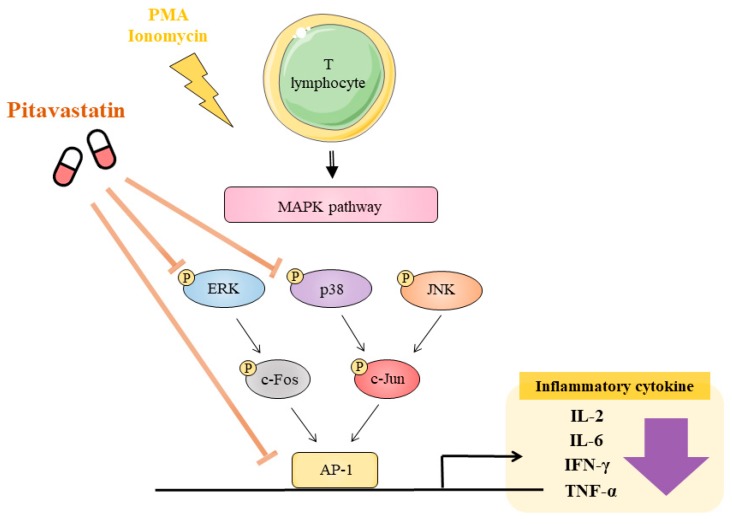
Schematic diagram of the anti-inflammation effects of pitavastatin in human primary T cells. Pitavastatin suppressed proinflammatory cytokine secretion through the inhibition of ERK/p38 MAPK pathways in PMA+ionomycin-activated human T cells and the reduction of transcriptional activity of AP-1 in PMA+ionomycin-activated 293T cells.

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
