# Peer review of "Pitavastatin Exerts Potent Anti-Inflammatory and Immunomodulatory Effects via the Suppression of AP-1 Signal Transduction in Human T Cells"

_ijms, 2019, doi:10.3390/ijms20143534_

Round 1
Reviewer 1 Report
Chen et al describe in this work the immunomodulatory effect of pitavastatin, a statin preparation, in human T cells. Their report first analyse how pitavastatin inhibits pro-inflammatory cytokine secretion profile (compared to non-treated and treated with other statin preparations) without affecting cell viability. This result is further confirmed by the quantification of cytokine mRNA. Furthermore, the authors identify that mechanistically, high doses of pitavastatin modestly suppresses the PMA+Ionomycin-induced phosphorylation of ERK and p38, while JNK and IKBa phosphorylation changes did not reach statistical significance upon statin treatment. The authors pinpoint AP-1 activity as a common readout of the signal transduction altered by pitavastatin, stating that the statin treatment reduces AP-1 nuclear content in T cells and transcriptional activity in 293T cells based on a reporter assay.
This work contributes to the evidence presented in several papers that have reported a pro-inflammatory role of statins preparations in in vitro culture of human mononuclear cells. In my opinion, the work is interesting and adequate for publication. However, I have both major and minor comments that should be addressed:
MAJOR COMMENTS
1. All the data presented in this work has undergone normalisation. This fact (along with a clear description of the data normalisation process, which is missing in some instances, see minor comments for specifics) should be very clearly addressed and explained in the text, figures and statistical analyses. The statistical comparisons should have been done in the raw data and this fact needs to be stated in the methods.
2. The data is show as mean +/- SEM. However, the methods do not describe if data adjust to normal distribution (which would be highly unlikely with n=3-5 per experiment). If these tests have been run and data adjust to normal, then this should be stated in the methods section. If these analyses have not been performed, then statistical descriptives more adequate to no-normal distributions should be used, like median and range. Personally, with such a low n, I would highly encouraged to show both median and individual data in all the graphs.
MINOR COMMENTS
1. The study compares the effect of pitavastatin to those of atorvastatin and ruvastatin, and show specificity of the pitavastatin treatment. Pitavastatin and atorvastatin are lipophilic statin preparations, that cross well cellular membranes. However, ruvastatin is a lipophobic statin preparation. The fact that authors do not show differences between atorvastatin and ruvastatin seems to suggest that some of the effects they describe might be ascribed to off-target effect of the pitavastatin. An easy way to address this would be to supplement the inhibitory treatment with mevalonate and expect recovery of the signal. As an alternative to these experiments, authors should at least discuss this effect.
2. Introduction: the main findings of this work apply to the immunomodulatory effects of pitavastatin in human T cells derived from healthy individuals, despite of the introduction being mainly focused on the role of statins and inflammation in cardiovascular disease. However, there are many reports that have addressed the role of different statin preparations in in vitro culture of human cells. Most of these studies have shown anti-inflammatory properties, although this is not consistent and still an area of intense research. Authors should include a few more of these studies (only one is cited, and not in healthy individuals’ mononuclear cells), to give a better overview of the current state of this area of research and place their findings into context.
3. Page 2, line 34. Heading of results does not reflect the data: why is the pitavastatin effect named as beneficial? This statement is more adequate for discussion in my opinion.
4. Page 2, line 47. I think the authors mean 10uM were tested (and not treated).
5. Page 2, line 50, methods and figure legends 1 and 2. The concentration of atorvastatin and ruvastatin used for these experiments is not noted anywhere. Hence, the statement “which are superior to atorvastatin and rosuvastatin” should be taken into the context of the dose (is it superior or not?).
6. Page 4, line 12. Cells are not “PMA + ionomycin – induced”, but activated, maybe?
7. Page 5, line 16. Authors state “transcriptional activity of AP-1 in activated T cells”, this in incorrect, as the experiment was performed in 293T cells (unless figure legend 4 is incorrect).
8. Page 6, line 16. Authors state that “the underlying mechanism is through inhibiting AP-1 transcriptional activity”.
9. Page 7, line 16. Extra space after ldlr-/-.
10. Page 8, line 6. “this finding also consistence”, I think the authors mean “this finding is also consistent”.
11. Page 8, line 32. Authors state that “pitavastatin targets AP-1 DNA binding and transcriptional activity in stimulated T cells”. This is incorrect, as the authors have shown that is the case for 293T, but not for T cells.
12. Page 8, line 43. “macrophages”.
13. Page 9, line 11. Methods state b-actin, but figures show tubulin.
14. Page 9, culture section. Dosage of PMA and ionomycin is missing. This section contains very little explanation about the culture conditions that should be improved (culture media, supplements, cell concentration, etc).
15. Page 9, line 18. PBMCs were (not was).
16. Page 9, line 26. Acetate (not acctate).
17. Page 10. ELISA methods: data normalisation process is not explained. This should also be stated in the corresponding figure legend.
18. Page 10. PCR methods: primers’ sequences should be supplied, as a table or within the text.
19. Page 9, line 29. Extra space in 14000.
20. Page 11, Luciferase assay. Data normalisation process needs an explanation. Was renilla used as internal control?
21. Comments to figure 1. “V” is not defined anywhere in the figure (vehicle control?); dose for PMA, ionomycin, atorvastatin and rosuvastatin should be stated. Symbol **** is not defined in the text, symbol # defines significant “versus the cell basal level”, what does this mean exactly? It seems to suggest that cells cultured for 24h without PMA+Io stimulation and drug have a significant lower amount of cytokine when compared to ex vivo? This is really confusing and I am unsure of the result the authors want to express. The text does not help either, as there is no reference to such a difference. Data normalisation should be explained.
22. Comments to figure 2. Line 2 “de novo”; V should be defined, data normalisation process should be explained. Graphs state protein names instead of mRNA species names. # is not clearly defined.
23. Comments to figure 3. Graphs on sections B and D have really poor quality, y-axes are nearly unreadable. “V is not defined. Data normalisation should be explained. * is not defined.
24. Comments to figure 4. Legend stated that (A) is representative of 3 independent experiments. As per rest of the manuscript, where representative and cumulative data has been shown, I think the same should be applied here for consistency, especially because the difference does not seem to be so obvious. **** is not defined.
25. In most cases across the whole manuscript ionomycin is misspelled.
26. There are inconsistencies among naming the drugs with capitals or without. I suggest to name the drugs without capital all across the manuscript.
Author Response
#Response to Reviewer 1
MAJOR COMMENTS
1. All the data presented in this work has undergone normalisation. This fact (along with a clear description of the data normalisation process, which is missing in some instances, see minor comments for specifics) should be very clearly addressed and explained in the text, figures and statistical analyses. The statistical comparisons should have been done in the raw data and this fact needs to be stated in the methods.
Reply: Thank you very much for the valuable comments. We reanalyzed our data with the raw data without normalization. The details regarding the statistical analyses have been provided in sections 4.10, page 11, line 33. All the figure legends with clear labeling and entire information have been checked.
2. The data is show as mean +/- SEM. However, the methods do not describe if data adjust to normal distribution (which would be highly unlikely with n=3-5 per experiment). If these tests have been run and data adjust to normal, then this should be stated in the methods section. If these analyses have not been performed, then statistical descriptives more adequate to no-normal distributions should be used, like median and range. Personally, with such a low n, I would highly encouraged to show both median and individual data in all the graphs.
Reply: Thank you for the suggestions. All the data have been represented with the median and individual data.
MINOR COMMENTS
1. The study compares the effect of pitavastatin to those of atorvastatin and rosuvastatin and show specificity of the pitavastatin treatment. Pitavastatin and atorvastatin are lipophilic statin preparations, that cross well cellular membranes. However, rosuvastatin is a lipophobic statin preparation. The fact that authors do not show differences between atorvastatin and rosuvastatin seems to suggest that some of the effects they describe might be ascribed to off-target effect of the pitavastatin. An easy way to address this would be to supplement the inhibitory treatment with mevalonate and expect recovery of the signal. As an alternative to these experiments, authors should at least discuss this effect.
Reply: Thank you very much for your comments. Different from the lipophilic nature of atorvastatin and pitavastatin, the lipophobic rosuvastatin performs intracellular effects via membrane bound transporter, such as organic anion transporting polypeptides (OATP), which does exist in T cells [1, 2]. Moreover, study from Nakagomi et al suggested that atorvastatin and pitavastatin exhibit distinct effects on inflammation in patients with dyslipidemia [3]. Such evidence partly explains the similar effects of atorvastatin and rosuvastatin on proinflammatory cytokines production in activated T cells, and further supports the specific pleiotropic effects of pitavastatin. The above description had been added in the discussion section (Page 8, line 11).
References
1. Kitamura, S., et al., Involvement of multiple transporters in the hepatobiliary transport of rosuvastatin. Drug Metab Dispos, 2008. 36(10): p. 2014-23.
2. Janneh, O., et al., Cultured CD4T cells and primary human lymphocytes express hOATPs: intracellular accumulation of saquinavir and lopinavir. Br J Pharmacol, 2008. 155(6): p. 875-83.
3. Nakagomi, A., et al., Differential Effects of Atorvastatin and Pitavastatin on Inflammation, Insulin Resistance, and the Carotid Intima-Media Thickness in Patients with Dyslipidemia. J Atheroscler Thromb, 2015. 22(11): p. 1158-71.
2. Introduction: the main findings of this work apply to the immunomodulatory effects of pitavastatin in human T cells derived from healthy individuals, despite of the introduction being mainly focused on the role of statins and inflammation in cardiovascular disease. However, there are many reports that have addressed the role of different statin preparations in in vitro culture of human cells. Most of these studies have shown anti-inflammatory properties, although this is not consistent and still an area of intense research. Authors should include a few more of these studies (only one is cited, and not in healthy individuals’ mononuclear cells), to give a better overview of the current state of this area of research and place their findings into context.
Reply: Thank you for your suggestion. Several studies pointed out the anti-inflammatory properties of statins. Statins display inhibitory effects on the release of pro-inflammatory cytokines, MCP-1, IL-6, and IL-8, and suppress matrix metalloproteinases, in both macrophages and SMCs [1, 2]. Specifically, pitavastatin and atorvastatin were proposed to exhibit inhibitory effects on apolipoprotein C3-induced vascular cell adhesion protein 1 (VCAM-1) expression in ECs and monocyte adhesion [3]. Regarding the immunomodulatory effects of statins in human T cells, statins repress oxidized LDL-induced dendritic cells (DC) maturation and suppress T cell activation with reduction of inflammatory cytokines production in T cells from carotid plaques or healthy individuals [4]. Additionally, statins are known to inhibit the differentiation of pro-inflammatory IL-17 helper T cells and promote forkhead box (FOX) P3+ CD4+ regulatory T cells, which is involved in atherosclerotic plaque stability [5]. In healthy human T cells, statins reduce T cell proliferation and IFN-γ production [6]. These studies highlighted the atheroprotective effects of statins in addition to their lipid-lowing properties. The above descriptions have been provided in the introduction section (page 2, line 12, and line 27).
References
1. Rezaie-Majd, A., et al., Simvastatin reduces expression of cytokines interleukin-6, interleukin-8, and monocyte chemoattractant protein-1 in circulating monocytes from hypercholesterolemic patients. Arterioscler Thromb Vasc Biol, 2002. 22(7): p. 1194-9.
2. Luan, Z., A.J. Chase, and A.C. Newby, Statins inhibit secretion of metalloproteinases-1, -2, -3, and -9 from vascular smooth muscle cells and macrophages. Arterioscler Thromb Vasc Biol, 2003. 23(5): p. 769-75.
3. Zheng, C., et al., Statins suppress apolipoprotein CIII-induced vascular endothelial cell activation and monocyte adhesion. Eur Heart J, 2013. 34(8): p. 615-24.
4. Frostegard, J., et al., Oxidized Low-Density Lipoprotein (OxLDL)-Treated Dendritic Cells Promote Activation of T Cells in Human Atherosclerotic Plaque and Blood, Which Is Repressed by Statins: microRNA let-7c Is Integral to the Effect. J Am Heart Assoc, 2016. 5(9): p. e003976.
5. Kim, Y.C., K.K. Kim, and E.M. Shevach, Simvastatin induces Foxp3+ T regulatory cells by modulation of transforming growth factor-beta signal transduction. Immunology, 2010. 130(4): p. 484-93.
6. Bu, D.X., et al., Statin-induced Kruppel-like factor 2 expression in human and mouse T cells reduces inflammatory and pathogenic responses. J Clin Invest, 2010. 120(6): p. 1961-70.
3. Page 2, line 34. Heading of results does not reflect the data: why is the pitavastatin effect named as beneficial? This statement is more adequate for discussion in my opinion.
Reply: Thank you for the comments. the heading of results 2.1 has been revised (page 2, line 41).
4. Page 2, line 47. I think the authors mean 10uM were tested (and not treated).
Reply: It has been revised (page 3, line 3).
5. Page 2, line 50, methods and figure legends 1 and 2. The concentration of atorvastatin and rosuvastatin used for these experiments is not noted anywhere. Hence, the statement “which are superior to atorvastatin and rosuvastatin” should be taken into the context of the dose (is it superior or not?).
Reply: Thank you for the reminding. The concentration of atorvastatin and rosuvastatin used in the experiments was 10 μM, which is the highest dose of pitavastatin used in the study. To avoid misunderstanding, we revised the labeling and the figure legend in figure 1 and 2. The effects of pitavastatin, atorvastatin, and rosuvastatin on the inhibition of PMA+ionomycin-induced cytokine production in T cells have been showed in figure 1 and 2.
6. Page 4, line 12. Cells are not “PMA + ionomycin – induced”, but activated, maybe?
Reply: Thank you for the suggestion. The term has been checked and revised in the whole manuscript.
7. Page 5, line 16. Authors state “transcriptional activity of AP-1 in activated T cells”, this in incorrect, as the experiment was performed in 293T cells (unless figure legend 4 is incorrect).
Reply: Thank you for the reminding. In fig 4, the EMSA was conducted in primary T cells and the luciferase assay was evaluated in 293T cells (page 5, line 20).
8. Page 6, line 16. Authors state that “the underlying mechanism is through inhibiting AP-1 transcriptional activity”.
Reply: Thank you for the comments. The texts have been revised in page 6, line 20.
9. Page 7, line 16. Extra space after ldlr-/-.
Reply: Thank you for the reminding. The texts have been revised in page 7, line 16.
10. Page 8, line 6. “this finding also consistence”, I think the authors mean “this finding is also consistent”.
Reply: Thank you for the reminding. The texts have been revised in page 8, line 6.
11. Page 8, line 32. Authors state that “pitavastatin targets AP-1 DNA binding and transcriptional activity in stimulated T cells”. This is incorrect, as the authors have shown that is the case for 293T, but not for T cells.
Reply: Thank you for the reminding. The results of AP-1 DNA binding and transcriptional activity were examined in primary T cells and 293T cells, respectively. The texts have been revised in page 8, line 37.
12. Page 8, line 43. “macrophages”.
Reply: Thank you for the reminding. The texts have been revised in page 8, line 39.
13. Page 9, line 11. Methods state b-actin, but figures show tubulin.
Reply: Thank you for the reminding. We used tubulin as the internal control in Western blotting, which has been revised in the method section (Page 9, line 21).
14. Page 9, culture section. Dosage of PMA and ionomycin is missing. This section contains very little explanation about the culture conditions that should be improved (culture media, supplements, cell concentration, etc).
Reply: Thank you for the comments. The dosage of PMA and ionomycin were 5 ng/mL and 1 μM, respectively. We revised the descriptions in page 9, line 37. Additionally, the information regarding cell culture conditions has been added in section 4.2, page 9, line 35.
15. Page 9, line 18. PBMCs were (not was).
Reply: Thank you for the reminding. The texts have been revised in page 9, line 26.
16. Page 9, line 26. Acetate (not acctate).
Reply: Thank you for the reminding. The texts have been revised in page 9, line 35.
17. Page 10. ELISA methods: data normalisation process is not explained. This should also be stated in the corresponding figure legend.
Reply: Thank you for the suggestion. We reanalyzed our data with the raw data without normalization. The figure legends have been revised.
18. Page 10. PCR methods: primers’ sequences should be supplied, as a table or within the text.
Reply: Thank you for the comments. The primers’ sequences used in this study have been listed in sections 4.5, page.10, line 30.
19. Page 9, line 29. Extra space in 14000.
Reply: Thank you for the reminding. The texts have been revised in page 11, line 9.
20. Page 11, Luciferase assay. Data normalisation process needs an explanation. Was renilla used as internal control?
Reply: Thank you for the suggestion. As indicated previously, we reanalyzed our data with raw data. Renilla was not used in our luciferase experiment. We did the single transfection with luciferase plasmids and seeded the transfected cells equally to make sure the same transfection efficiency. The above description had been added in the method section (Page 11, line 29)
21. Comments to figure 1. “V” is not defined anywhere in the figure (vehicle control?); dose for PMA, ionomycin, atorvastatin and rosuvastatin should be stated. Symbol **** is not defined in the text, symbol # defines significant “versus the cell basal level”, what does this mean exactly? It seems to suggest that cells cultured for 24h without PMA+Io stimulation and drug have a significant lower amount of cytokine when compared to ex vivo? This is really confusing and I am unsure of the result the authors want to express. The text does not help either, as there is no reference to such a difference. Data normalisation should be explained.
Reply: Thank you for the suggestion. The figures and figure legends have been revised carefully. The symbols have been indicated clearly. To avoid misunderstanding, we used the asterisk to represent the significance. The group difference was analyzed between the indicated groups and the PMA+ionomycin-stimulated group.
22. Comments to figure 2. line 2 “de novo”; V should be defined, data normalisation process should be explained. Graphs state protein names instead of mRNA species names. # is not clearly defined.
Reply: Thank you for the suggestion. The figure legends have been revised carefully (Page 4, line 11). The labeling of mRNA expressions have been revised.
23. Comments to figure 3. Graphs on sections B and D have really poor quality, y-axes are nearly unreadable. “V is not defined. Data normalisation should be explained. * is not defined.
Reply: Thank you for the suggestion. The graphs on section B and D have been revised. The figure legends with the detail information have been provided (Page 5, line 13)
24. Comments to figure 4. Legend stated that (A) is representative of 3 independent experiments. As per rest of the manuscript, where representative and cumulative data has been shown, I think the same should be applied here for consistency, especially because the difference does not seem to be so obvious. **** is not defined.
Reply: Thank you for the suggestion. The results of EMSA from 3 independent donors were quantified and analyzed. The figure legends with the detail information have been provided (Page 6, line 10)
25. In most cases across the whole manuscript ionomycin is misspelled.
Reply: Thank you for the reminding. We corrected the misspelling in all manuscripts.
26. There are inconsistencies among naming the drugs with capitals or without. I suggest to name the drugs without capital all across the manuscript.
Reply: Thank you for the reminding. We revised the name of the drugs without capitals.
Reviewer 2 Report
The major finding is reported to be that pitavastatin is better at inhibiting a direct effect on T cells as compared to 2 other statins. There is not enough data to support this conclusion. There are no concentrations given for the other statins, and it could be that they have similar effects but at somewhat different concentrations. The authors must clearly show that the other statins, with conc curves are not as good as pitavastatin.
The concept in general is of interest, since statins may very well be immunomodulatory also in the clinic.
Previous studies on immune modulatory properties are not mentioned, for example Liu et al, JAHA 2016 where an immune modulatory effect on T cell activation was for the first time reported in human plaque cells.
Author Response
#Response to Reviewer 2
Comments and Suggestions for Authors
The major finding is reported to be that pitavastatin is better at inhibiting a direct effect on T cells as compared to 2 other statins. There is not enough data to support this conclusion. There are no concentrations given for the other statins, and it could be that they have similar effects but at somewhat different concentrations. The authors must clearly show that the other statins, with conc curves are not as good as pitavastatin.
Reply: Thank you for the constructive comments. The concentration of atorvastatin and rosuvastatin used in the experiments was 10 μM, which is the highest dose of pitavastatin used in the study. To avoid misunderstanding, we revised the labeling and the figure legend in figure 1 and 2. The effects of pitavastatin, atorvastatin, and rosuvastatin on the inhibition of PMA+ionomycin-induced cytokine production in T cells have been shown in figure 1 and 2.
The concept in general is of interest, since statins may very well be immunomodulatory also in the clinic.
Reply: Thank you for your comments.
Previous studies on immune modulatory properties are not mentioned, for example Liu et al, JAHA 2016 where an immune modulatory effect on T cell activation was for the first time reported in human plaque cells.
Reply: Thank you for your comments. Liu et al demonstrated that statins repress oxidized LDL-induced dendritic cells (DC) maturation and suppress T cell activation with reduction of inflammatory cytokines production in T cells from carotid plaques or healthy individuals [1]. The above description had been added in the introduction section (page 2, line 27).
References
1. Kim, Y.C., K.K. Kim, and E.M. Shevach, Simvastatin induces Foxp3+ T regulatory cells by modulation of transforming growth factor-beta signal transduction. Immunology, 2010. 130(4): p. 484-93.